# Cross-Subject Emotion Recognition with CT-ELCAN: Leveraging Cross-Modal Transformer and Enhanced Learning-Classify Adversarial Network [note 1]

**DOI:** 10.3390/bioengineering12050528

**Published:** 2025-05-15

**Authors:** Ping Li, Ao Li, Xinhui Li, Zhao Lv

**Affiliations:** 1School of Computer Science and Technology, Anhui University, Hefei 230601, China; pingli0112@gmail.com (P.L.); la@stu.ahu.edu.cn (A.L.); xinhuili@ahu.edu.cn (X.L.); 2The Key Laboratory of Flight Techniques and Flight Safety, Civil Aviation Flight University of China, Deyang 618307, China

**Keywords:** multimodal emotion recognition, cross-modal transformer, adversarial learning, cross-subject generalization, physiological signals

## Abstract

Multimodal physiological emotion recognition is challenged by modality heterogeneity and inter-subject variability, which hinder model generalization and robustness. To address these issues, this paper proposes a new framework, Cross-modal Transformer with Enhanced Learning-Classifying Adversarial Network (CT-ELCAN). The core idea of CT-ELCAN is to shift the focus from conventional signal fusion to the alignment of modality- and subject-invariant emotional representations. By combining a cross-modal Transformer with ELCAN, a feature alignment module using adversarial training, CT-ELCAN learns modality- and subject-invariant emotional representations. Experimental results on the public datasets DEAP and WESAD demonstrate that CT-ELCAN achieves accuracy improvements of approximately 7% and 5%, respectively, compared to state-of-the-art models, while also exhibiting enhanced robustness.

## 1. Introduction

With growing interest in embodied intelligence [1], emotion recognition has gained renewed attention as a core mechanism for enabling machines to perceive and respond to human affective states [2]. It supports a range of applications, from personalized education [3] to emotion-aware intelligent systems [4], ultimately contributing to more human-centered technologies [5]. Most existing emotion recognition systems rely on non-physiological cues such as facial expressions, speech, or text [6]. While effective in controlled environments, these signals are often susceptible to environmental noise, voluntary masking, and cultural variability, limiting their robustness and generalizability in real-world scenarios. In contrast, physiological signals—such as EEG and EDA—offer a direct, unconscious reflection of emotional states, and demonstrate advantages in real-time responsiveness, data richness, and sensitivity to individual differences [7], making them promising for emotion-aware systems in practice [8,9].

However, most existing physiological approaches rely on single-modal signals, which are insufficient to capture the complexity and dynamics of human emotions [10,11]. Multimodal physiological emotion recognition has thus gained increasing attention for its potential to leverage complementary information across modalities, enhancing recognition accuracy and improving generalization, especially in cross-subject scenarios. However, naive fusion strategies often introduce alignment mismatches, redundant information, and noise-induced distortions, ultimately degrading system performance [12].

Recent progress in deep learning, particularly the development of Transformer architectures, has introduced new opportunities for multimodal physiological emotion recognition. With powerful capabilities in feature extraction and interaction modeling, Transformers effectively capture complex dependencies across modalities and have demonstrated significant improvements in recognition accuracy [13,14,15]. Among these efforts, Wang et al. [16] proposed the Cross-modal Transformer (CT), a representative architecture that models both intra- and inter-modal relationships to enhance feature representation. However, CT does not explicitly address inter-subject variability—where physiological responses to identical emotional stimuli may differ substantially across individuals—limiting its generalization in cross-subject scenarios, which remains a key challenge in physiological emotion recognition.

To address the limitations of existing cross-modal models in handling subject variability and semantic misalignment, we propose CT-ELCAN, a novel end-to-end architecture designed to learn emotionally meaningful, modality-invariant, and subject-independent representations. The core idea of CT-ELCAN is to shift the focus from direct multimodal feature fusion to the alignment of emotionally meaningful, modality-invariant, and subject-independent representations. This perspective treats multimodal emotion recognition as a problem of semantic-level alignment across heterogeneous physiological inputs and individual differences, rather than a simple signal integration task.

CT-ELCAN consists of two major components: a cross-modal Transformer module and an Enhanced Learning-Classifying Adversarial Network (ELCAN). While the Transformer is responsible for modeling contextual interactions across heterogeneous physiological modalities, ELCAN plays the central role in improving model robustness and generalization. ELCAN is designed to extract emotion-discriminative yet subject-invariant representations through adversarial alignment, aligning feature distributions at the semantic level while preserving classification performance. Together, both modules enable CT-ELCAN to effectively capture shared emotional representations while resisting overfitting and identity bias. Despite its integrated design, CT-ELCAN must address the following technical challenges:**Multimodal feature heterogeneity.** This challenge refers to the difficulty of effectively integrating features derived from different physiological modalities. It arises from inconsistencies in sampling rates, temporal resolution, and data format across modalities. Such heterogeneity disrupts feature compatibility and hampers the model’s ability to extract semantically aligned emotional representations.**Inter-subject variability.** This challenge involves the inconsistency of emotional patterns across individuals, even under the same stimulus conditions. It stems from individual differences in physiological structure, emotional expression style, and sensor sensitivity. These variations hinder the model’s ability to learn generalized representations, limiting its performance on unseen subjects.**Adversarial training instability.** This challenge reflects the inherent tension between learning features that are both emotion-discriminative and subject-invariant. It emerges from the conflicting optimization goals of the emotion classifier and the subject discriminator during adversarial training. Such conflict can destabilize gradient updates, leading to convergence difficulties and degraded performance.

CT-ELCAN addresses these challenges through three targeted strategies. Firstly, the cross-modal Transformer employs modality-aware attention mechanisms to reduce inter-modality discrepancies and improve semantic alignment across heterogeneous features. Secondly, adversarial learning in ELCAN is applied to minimize subject-related variance while maintaining emotion-discriminative capability, thus enhancing the generalization across individuals. Lastly, to mitigate the instability of adversarial training, CT-ELCAN integrates two complementary mechanisms: a Gradient Reversal Layer (GRL) combined with a triplet adversarial loss to balance competing optimization objectives, and a conditional Signal-Adaptive GAN (c-SAGAN) to generate subject-aware augmented samples that improve distributional robustness and training stability.

The main contributions of this paper are summarized as follows:We propose ELCAN, an enhanced adversarial learning-classifying network that integrates gradient-based feature alignment and conditional data augmentation. By jointly optimizing classification and invariance objectives, ELCAN effectively learns modality-invariant and subject-independent emotional representations, thereby improving generalization across subjects and modalities.Based on ELCAN, we develop CT-ELCAN, an end-to-end multimodal emotion recognition framework that combines a cross-modal Transformer for contextual feature modeling with ELCAN for robust, invariant representation learning. This unified design enables effective semantic alignment across heterogeneous signals while maintaining emotional discriminability.

The remainder of this paper is organized as follows: Section 2 reviews the related work; Section 3 introduces the preliminary knowledge; Section 4 presents the detailed architecture and training strategy of CT-ELCAN; Section 5 reports the experimental results and performance comparison; we discuss some limitation of CT-ELCAN in Section 6 and concludes this paper in Section 7.

This article is a revised and expanded version of a paper entitled “CAT-LCAN: A Multimodal Physiological Signal Fusion Framework for Emotion Recognition [17]”, which was presented at BICS 2024 (The 14th International Conference on Advances in Brain Inspired Cognitive Systems (Heifei, Anhui, China in 6 December 2024–28 January 2025)).

## 2. Related Work

### 2.1. Affective Representations

Emotion modeling provides the theoretical basis for emotion recognition tasks. Among the most influential paradigms are the discrete emotion model and the dimensional emotion model. The discrete model defines a fixed set of universal emotions—such as happiness, anger, sadness, fear, and surprise—rooted in biological expression systems [18]. Although widely used in early studies, it is limited in representing subtle affective variations.

To address this, dimensional models represent emotional states in continuous spaces, commonly using valence and arousal dimensions [19]. Such representations offer a more flexible framework for capturing emotional transitions and ambiguity, and are particularly suitable for analyzing physiological signals. Studies have shown that mapping physiological data to a valence–arousal space facilitates robust emotion representation and model training [20]. Accordingly, this work adopts a dimensional emotion model to support the design of our recognition framework.

### 2.2. Physiological Emotion Recognition

Physiological signals such as electroencephalogram (EEG), electromyography (EMG), electrooculography (EOG), and galvanic skin response (GSR) have gained prominence in affective computing due to their ability to reflect involuntary and spontaneous emotional responses [7,21]. These biosignals are less susceptible to voluntary manipulation and environmental interference, offering advantages over facial and speech-based cues in emotion recognition.

Single-modality physiological emotion recognition has achieved promising results by extracting handcrafted or frequency-domain features from individual signal types [22,23]. Jenke et al. [24] conducted a systematic evaluation of EEG-based feature extraction methods and demonstrated the effectiveness of frequency-band energy features in classifying affective states. Similarly, Koelstra et al. [25] introduced the DEAP dataset, which provides synchronized EEG and peripheral physiological recordings annotated along valence–arousal dimensions, laying the groundwork for multimodal affective computing research.

Despite these advances, single-modality systems often struggle to fully capture the richness and complexity of human emotion. More critically, physiological responses vary significantly across individuals due to differences in anatomical structure, sensor placement, and emotional sensitivity. This inter-subject variability severely limits the generalization capability of models trained on subject-dependent distributions, especially in real-world cross-subject applications [26]. These limitations have motivated recent efforts toward multimodal fusion and subject-invariant representation learning.

### 2.3. Deep Multimodal Fusion Methods

Deep learning has led to considerable progress in multimodal physiological emotion recognition by enabling more effective representation learning and modality integration. Fusion methods are typically divided into two paradigms: joint representation learning and coordinated representation learning [27].

Joint learning methods combine features from different modalities into a shared latent space. Liu et al. [28] proposed a bimodal deep autoencoder to fuse EEG features with other physiological signals for emotion classification. Tang et al. [29] utilized a bimodal LSTM network to extract temporal and frequency-domain patterns jointly from multimodal data.

In contrast, coordinated fusion strategies maintain separate modality-specific representations and align them through statistical or semantic constraints. For example, DCCA was used by Liu et al. [30] to maximize correlation between modalities while preserving emotion-related features. Tang et al. [31] developed RHPRNet, a hierarchical fusion network that combines EEG and peripheral features with global attention modules to improve cross-domain emotional pattern learning.

However, few of these approaches explicitly tackle inter-subject variability or training instability, both of which are critical challenges in cross-subject physiological recognition. These limitations motivate our work, which unifies multimodal attention, adversarial alignment, and augmentation-based regularization.

## 3. Preliminary

### 3.1. Cross-Modal Transformer

The Cross-modal Transformer (CT) is a self-attention-based architecture designed to capture semantic interactions across heterogeneous modalities. It dynamically models both modality-specific and cross-modal dependencies, making it well-suited for tasks such as physiological emotion recognition where signals are diverse and temporally misaligned [16].

Figure 1 illustrates the overall structure of CT as adopted in our framework. The architecture consists of four main components: a modality-specific projection layer, sinusoidal positional encoding, a cross-modal attention block, and stacked encoder layers for hierarchical representation learning.

Given an input sequence $X_{m_{i}}\in R^{L,D_{i}}$ from the *i*-th modality, where *L* is the temporal length and $D_{i}$ is the channel dimension, a 1D pointwise convolution is first applied to project the input into a shared feature space of dimension *D*:(1)$${\bar{X}}_{m_{i}}=\mathrm{Conv}1D\left(X_{m_{i}},1\right)$$

This results in the unified feature representation ${\bar{X}}_{m_{i}}\in R^{L,D}$.

To incorporate temporal order, sinusoidal positional encodings are added to the projected features. The encoding matrix $PE\in R^{L,D}$ is defined as:(2)$$\begin{array}{cc} PE[pos,2k] & =\sin\left(\frac{pos}{10000^{2k/D}}\right),\\ PE[pos,2k+1] & =\cos\left(\frac{pos}{10000^{2k/D}}\right) \end{array}$$
where pos∈{1,…,L} denotes the temporal position index, and *k* indexes the feature dimension. The position-aware representation is computed as ${\bar{X}}_{m_{i}}^{p}={\bar{X}}_{m_{i}}+PE$.

After positional encoding, features from all modalities are concatenated along the temporal axis to construct the global multimodal context ${\bar{X}}_{F}\in R^{L_{T},D}$, where $L_{T}$ is the combined sequence length across modalities. Subsequently, a cross-modal attention module is introduced to capture semantic relationships between each unimodal sequence and the global context. Each unimodal sequence ${\bar{X}}_{m_{i}}^{p}$ is transformed into a query matrix, while the global multimodal context provides the key and value:(3)$$\begin{array}{cc} Q & ={\bar{X}}_{m_{i}}^{p}W_{Q},\;K={\bar{X}}_{F}W_{K},\;V={\bar{X}}_{F}W_{V} \end{array}$$

Here, $W_{Q}$, $W_{K}$, and $W_{V}\in R^{D,D^{\mathrm{attn}}}$ are learnable projection matrices that map input features to the attention subspace of dimension $D^{\mathrm{attn}}$. The resulting matrices *Q*, *K*, and $V\in R^{L,D^{\mathrm{attn}}}$ represent the query, key, and value, respectively. Cross-modal alignment is then computed via scaled dot-product attention:(4)$$\mathrm{Attention}\left(Q,K,V\right)=\mathrm{softmax}\left(\frac{QK^{⊤}}{\sqrt{D^{\mathrm{attn}}}}\right)V$$

This allows each modality to selectively attend to semantically relevant segments from the overall multimodal input.

To further enhance contextual modeling, CT stacks *U* identical encoder layers, each comprising multi-head attention, feed-forward submodules, residual connections, and layer normalization. The final output for modality *i* is obtained as:(5)$$Y_{m_{i}}=\mathrm{CTEncoder}\left({\bar{X}}_{m_{i}}\right)$$
where $Y_{m_{i}}\in R^{L,D}$ denotes the refined, context-enriched representation that captures both intra- and inter-modal dependencies.

By capturing both modality-specific and cross-modal contextual features, these outputs form a unified embedding space that supports subsequent alignment and classification stages.

### 3.2. Self-Attention Generative Adversarial Network (SAGAN)

The Self-Attention Generative Adversarial Network (SAGAN) enhances conventional GANs by integrating a self-attention mechanism into the generator and discriminator. This design allows the model to capture long-range dependencies and global contextual structures, which are particularly beneficial in scenarios involving complex and high-dimensional signals. The underlying adversarial learning objective remains the same as in standard GANs:(6)$$\min_{\theta_{G}}\max_{\theta_{D}}L\left(X_{r},X_{g}\right)=E_{x_{r}∼X_{r}}\left[\log D\left(x_{r}\right)\right]+E_{x_{g}∼X_{g}}\left[\log\left(1-D\left(x_{g}\right)\right)\right]$$
where $x_{g}=G\left(x_{z}\right)$ is a generated sample based on a noise vector $x_{z}$, and $x_{r}$ is drawn from the real data distribution $X_{r}$. $\theta_{G}$ and $\theta_{D}$ denote the parameters of the generator and discriminator, respectively.

The self-attention mechanism in SAGAN operates on intermediate feature maps $x\in R^{C\times N}$, where *C* represents the number of channels and *N* denotes the number of spatial or temporal positions. The attention weights are calculated as:(7)$$\beta_{j,i}=\frac{\exp(s_{ij})}{\sum_{i=1}^{N}\exp\left(s_{ij}\right)},\;s_{ij}=f{\left(x_{i}\right)}^{⊤}g\left(x_{j}\right)$$

Here, $f\left(x\right)=W_{f}x$ and $g\left(x\right)=W_{g}x$ are learnable transformations, and $\beta_{j,i}$ measures how much the *j*-th position attends to the *i*-th one. The output of the attention layer is computed as:(8)$$O_{j}=v\left(\sum_{i=1}^{N}\beta_{j,i}h\left(x_{i}\right)\right),\;h\left(x_{i}\right)=W_{h}x_{i},\;v\left(x\right)=W_{v}x$$
with $W_{f}$, $W_{g}$, $W_{h}$, $W_{v}\in R^{\bar{C}\times C}$ being learnable projection matrices. The final output incorporates residual learning:(9)$$y_{i}=\gamma o_{i}+x_{i}$$
where γ is a learnable scalar initialized to zero.

Although SAGAN improves feature representation through global context modeling, its structure lacks semantic guidance during training. Specifically, the generator produces feature distributions without incorporating class-level information, which weakens the ability of the discriminator to learn category-aware boundaries. This class-agnostic generation process introduces noise into adversarial training, potentially distorting the feature space and reducing the effectiveness of cross-subject generalization.

## 4. Methodology

### 4.1. Algorithm Overview

This section presents the overall architecture and module designs of the proposed CT-ELCAN framework. CT-ELCAN is developed to address three key challenges in multimodal physiological emotion recognition: (i) heterogeneity across signal modalities, (ii) individual variability among subjects, and (iii) instability in adversarial training. As shown in Figure 2, the entire system consists of two main components: the Cross-modal Transformer (CT) and the Enhanced Learning-Classifying Adversarial Network (ELCAN).

In the first stage, raw physiological signals from different modalities (e.g., EEG, EOG, EMG, and GSR) are independently projected into a unified feature space via 1D pointwise convolution, followed by positional encoding. These signals are then processed by a multi-layer cross-modal Transformer encoder to obtain contextualized unimodal features and a fused multimodal representation. Benefiting from the cross-modal attention mechanism of the Transformer, each modality-specific feature captures not only its own long-term information but also integrates semantically aligned signals from other modalities. This facilitates cross-modal feature complementarity and improves the consistency of representations across modalities.

In the second stage, these enhanced modality-specific features are forwarded to the ELCAN module, which comprises two submodules: LCAN and c-SAGAN. LCAN performs adversarial training to encourage the learning of modality and subject-invariant features, thus enhancing model generalization across unseen individuals and conditions. Meanwhile, c-SAGAN introduces a category-aware adversarial generation mechanism to assist feature alignment and promote training stability. The full system is trained end-to-end with jointly optimized classification and adversarial objectives. The following subsections describe the individual components of CT-ELCAN in detail.

### 4.2. Cross-Modal Transformer Module

To address the heterogeneity of multimodal physiological signals, CT-ELCAN adopts a Cross-modal Transformer (CT) module as its backbone feature extractor. Since CT is a well-studied approach and has already been introduced in detail in Section 3, we do not revisit the underlying model here. Instead, we briefly outline its configuration and role within the CT-ELCAN framework.

As illustrated in the upper portion of Figure 2, each input modality (e.g., EEG, EOG, EMG, and GSR) is first projected into a unified feature space using one-dimensional pointwise convolution. Positional encodings are then added to preserve the temporal structure of the sequences. The resulting representations are processed by a Transformer encoder consisting of 3 layers, each with 4 attention heads and a hidden size of 128. This yields modality-specific features that are complementary, semantically enriched, and mutually informed through cross-modal attention, which are subsequently forwarded to the ELCAN module for adversarial alignment and downstream emotion recognition.

To enhance cross-subject generalization and mitigate overfitting, we propose a novel adversarial alignment module: the Enhanced Learning-Classifying Adversarial Network (ELCAN). As illustrated by the cyan-colored components in Figure 2, ELCAN is positioned downstream of the Cross-modal Transformer and consists of two sequentially connected sub-modules: c-SAGAN and LCAN. The c-SAGAN module performs conditional data augmentation by generating diverse, label-consistent training samples across modalities. Its outputs are then fed into the LCAN module, which applies adversarial training to extract modality-invariant and subject-invariant features. Together, these modules form an integrated pipeline that strengthens the generalization and robustness of the overall framework. In the following sections, we describe c-SAGAN and LCAN in the order they appear in the data processing pipeline.

#### c-SAGAN Submodule

To overcome the limitations of conventional emotional data generation methods, including insufficient sample diversity and inadequate alignment with target emotional categories, we propose a conditional self-attention GAN (c-SAGAN). The core idea of c-SAGAN is to enhance both the realism and label consistency of generated data by incorporating auxiliary emotional labels and self-attention mechanisms into the classical SAGAN framework. This design allows the model to better capture global dependencies and category-specific features within multimodal physiological signals.

As illustrated in Figure 3, the generator in c-SAGAN takes modality-specific feature maps *x* extracted from the Cross-modal Transformer as its primary input. During training, it also incorporates random noise *z* sampled from a prior distribution and an auxiliary emotional label $Y_{r}$ drawn from the target label space. These additional components guide the conditional generation process and ensure that the synthesized outputs are aligned with specific emotional categories. The discriminator then evaluates both real samples (x,y) and generated samples (G(z),y), assessing their authenticity and semantic consistency conditioned on $Y_{r}$.

### 4.3. ELCAN Module

To capture long-range dependencies within each modality, the generator integrates a self-attention mechanism. The input *x* is projected into three latent representations f(x), g(x), and h(x) using 1×1 convolutional layers. Here, f(x) and g(x) act as the query and key embeddings, respectively, and their inner product determines the attention score $s_{ij}$. These scores are normalized to compute the attention weights $\beta_{j,i}$, which determine the influence of the *i*-th feature on the *j*-th position. A weighted sum over h(x) is then computed and transformed by v(·), scaled by a learnable parameter γ, and added to the original input *x* via a residual connection. This process yields a globally contextualized and label-conditioned feature representation that is subsequently passed to the discriminator.

The model is trained using a hinge loss formulation. The discriminator aims to distinguish real from generated samples while ensuring that emotional semantics are preserved:(10)$$\begin{array}{cc} L_{D} & =-E_{\left(x,y\right)∼p_{\mathrm{data}}}\left[\min\left(0,-1+D\left(\left(x,y\right)|Y_{r}\right)\right)\right]\\ \; & \;-E_{z∼p_{z},y∼p_{\mathrm{data}}}\left[\min\left(0,-1-D\left(\left(G\left(z\right),y\right)|Y_{r}\right)\right)\right] \end{array}$$

The generator is optimized to fool the discriminator while maintaining label consistency:(11)$$\begin{array}{c} L_{G}=-E_{z∼p_{z},y∼p_{\mathrm{data}}}D\left(\left(G\left(z\right),y\right)|Y_{r}\right) \end{array}$$

By combining conditional adversarial training with global self-attention, c-SAGAN enhances the representational quality of synthesized data, supporting more robust and generalizable downstream emotion recognition.

#### LCAN Submodule

Following c-SAGAN, the Learning-Classifying Adversarial Network (LCAN) is employed to align cross-modal and cross-subject feature distributions. As illustrated in Figure 4, LCAN takes as input both the real multimodal features and the synthetic samples generated by c-SAGAN. Its objective is to learn modality-invariant and subject-independent representations that support robust emotion recognition.

LCAN consists of five functional components: the Feature Learning Module, Modality Classifier, Subject Classifier, Multimodal Fusion Module, and Emotion Classifier. The Feature Learning Module receives the enhanced intermediate unimodal features $Y_{M_{i}}$ and transforms them into high-level representations $Z_{M_{i}}\in R^{L,D}$ using a convolutional block:(12)$$Z_{M_{i}}=\mathrm{ConvBlock}\left(Y_{M_{i}}\right)$$
where ConvBlock(·) includes a sequence of convolution, normalization, activation, and pooling layers. Here, *L* denotes the temporal length of the feature sequence, and *D* is the feature dimension.

The outputs from all modalities are concatenated to form a multimodal representation $Z_{F}\in R^{L_{F},D}$:(13)$$Z_{F}=\mathrm{Concat}\left(Z_{M_{i}}\right),\;i=1,2,\ldots ,m$$
where $L_{F}$ denotes the total temporal length after concatenating all modalities.

To encourage modality- and subject-invariant representations, we use two auxiliary classifiers. The Modality Classifier receives $Z_{M_{i}}$ and predicts the modality label:(14)$$O_{M}=\mathrm{Softmax}\left(f_{\mathrm{mod}}\left(Z_{M_{i}}\right)\right),\;L_{\mathrm{mod}}=-\sum_{i=1}^{m}y_{i}\log\left(O_{M}\right)$$

The Subject Classifier takes $Z_{F}$ as input and predicts the subject label:(15)$$O_{S}=\mathrm{Softmax}\left(f_{\mathrm{sub}}\left(Z_{F}\right)\right),\;L_{\mathrm{sub}}=-\sum_{i=1}^{s}t_{i}\log\left(O_{S}\right)$$
where $f_{\mathrm{mod}}\left(\cdot \right)$ and $f_{\mathrm{sub}}\left(\cdot \right)$ denote the classifier mapping functions, and $y_{i}$ and $t_{i}$ are the true modality and subject labels. Here, $L_{\mathrm{mod}}$ and $L_{\mathrm{sub}}$ denote the cross-entropy losses for modality and subject classification, respectively, which are used in adversarial training to enforce modality- and subject-invariant representations.

The Feature Learning Module is trained adversarially against these classifiers. While the classifiers aim to minimize their respective losses, the feature extractor seeks to maximize them to eliminate modality- and subject-specific signals:(16)$$\min_{\theta_{\mathrm{mod}}}L_{\mathrm{mod}},\;\min_{\theta_{\mathrm{sub}}}L_{\mathrm{sub}},\;\max_{\theta_{\mathrm{feat}}}\left(L_{\mathrm{mod}}+L_{\mathrm{sub}}\right)$$

The fused multimodal feature $Z_{F}$ is further passed to the emotion classifier, composed of two fully connected layers:(17)$$O_{E}=\mathrm{Softmax}\left(f_{\mathrm{emo}}\left(Z_{F}\right)\right),\;L_{\mathrm{emo}}=-\sum_{i=1}^{n}k_{i}\log\left(O_{E}\right)$$
where $f_{\mathrm{emo}}\left(\cdot \right)$ is the mapping function of the classifier and $k_{i}$ is the ground-truth emotion label. Here, $L_{\mathrm{emo}}$ represents the standard cross-entropy loss for emotion prediction.

To align the opposing training objectives and enable joint optimization, a Gradient Reversal Layer (GRL) is introduced between the feature extractor and both auxiliary classifiers. During backpropagation, GRL inverts the gradients of adversarial losses, allowing the overall network to be trained end-to-end.

The final objective integrates all components:(18)$$L=\alpha L_{\mathrm{mod}}+\beta L_{\mathrm{sub}}+\gamma L_{\mathrm{emo}}+\phi L_{\mathrm{gan}}$$
where α, β, γ, and ϕ are scalar weights to balance the contributions of the respective losses. Here, $L_{\mathrm{gan}}$ is the combined hinge loss derived from the c-SAGAN generator and discriminator. All components are trained jointly in an end-to-end manner using the aggregated loss L.

Through this iterative adversarial training process, LCAN effectively disentangles modality and subject biases, yielding more robust and generalizable emotional representations for downstream recognition.

## 5. Performance Evaluation

### 5.1. Dataset and Experimental Setup

We evaluated the proposed CT-ELCAN model on two widely used public datasets of multimodal physiological signals: the DEAP dataset [25] and the WESAD dataset [32]. For the DEAP dataset, we performed binary classification tasks based on the valence and arousal dimensions of emotion. For the WESAD dataset, a three-class classification task was conducted to distinguish among neutral, stressed, and amused emotional states.

To ensure fairness and robustness in evaluation, we adopted a Leave-One-Subject-Out (LOSO) cross-validation strategy. In each iteration, one subject’s data were held out as the test set, while the remaining data were used for training. This process was repeated until every subject had been used once for testing, and the final performance was reported as the average accuracy across all subjects.

For model training, we adopted the Adam optimizer [33] with a fixed learning rate of $10^{-4}$, and applied gradient clipping when the gradient norm exceeded 10 to ensure training stability. All models were trained for 50 epochs with a batch size of 128 using the PyTorch (Version 2.0) framework [34], executed on a GeForce GTX 1080Ti GPU. Task-level parallelism was optimized to accommodate hardware constraints. To further improve performance across training rounds, we employed a grid search strategy to comprehensively explore the hyperparameter space and identify the optimal configuration that maximizes model performance. The final parameter settings are summarized in Table 1.

### 5.2. Experimental Results and Analysis

To evaluate the effectiveness and generalization ability of the proposed CT-ELCAN framework, we conducted comprehensive cross-subject comparative experiments under the Leave-One-Subject-Out (LOSO) cross-validation setting. We first compared CT-ELCAN with several widely adopted deep learning models, including CNN, LSTM, GRU, and Transformer. As shown in the upper part of Table 2, these classical architectures exhibit limited ability to handle modality heterogeneity and inter-subject variability. Among them, the Transformer model achieves the highest accuracy, benefiting from its attention mechanism; however, it still falls short in terms of generalization.

To further assess performance, we compared CT-ELCAN against recent state-of-the-art methods, including single-modal EEG-based models such as TARDGCN [35] and DSSN [36], as well as multimodal physiological signal-based models such as RDFKM [37], MCMT [38], and DGR-ERPS [39]. All models were trained and evaluated under identical experimental conditions to ensure fairness and comparability. The corresponding results are summarized in the lower part of Table 2.

As shown in Table 2, CT-ELCAN consistently outperforms all competing methods across four classification tasks. On the DEAP dataset, it achieves average accuracies of 70.82% and 71.34% for valence and arousal classification, respectively, surpassing the best-performing baseline RDFKM by over 7%. In the more challenging four-class classification task on DEAP, it also achieves the highest accuracy of 63.07%. On the WESAD dataset, CT-ELCAN reaches an accuracy of 73.93%, outperforming RDFKM by over 5%.

To better examine the model’s cross-subject generalization, Figure 5 visualizes both the individual-level accuracy distribution (left) and the corresponding cumulative distribution function (CDF, right) for each classification task. The bar plots on the left side reflect how the model performs on each subject, showing CT-ELCAN’s stability across diverse individuals. The CDF curves on the right help quantify this stability by showing the proportion of subjects achieving a given accuracy threshold.

Across all tasks, CT-ELCAN shows consistently better CDF performance. On the DEAP-Valence task, 50% of subjects achieve at least 70.82% accuracy, compared to 62% for RDFKM. At the 90% percentile, CT-ELCAN reaches 78%, whereas RDFKM only reaches 74%. On DEAP-Arousal, CT-ELCAN surpasses RDFKM by 7.3% at the 90% mark. On DEAP-Quaternary and WESAD tasks, the advantage at the 90% percentile is also evident, with CT-ELCAN outperforming RDFKM by around 3% to 5%. These results confirm that CT-ELCAN not only improves average accuracy but also provides more stable performance across subjects.

### 5.3. Ablation Experiments

To comprehensively assess the contribution of each core module in the CT-ELCAN framework—namely the Cross-modal Transformer (CT), the Conditional Self-Attention Generative Adversarial Network (c-SAGAN), and the Learning-Classifying Adversarial Network (LCAN)—we conducted a series of ablation experiments. These experiments aim to reveal the individual roles and effectiveness of each component in improving overall model performance.

We first evaluated the impact of the data augmentation module c-SAGAN by conducting two comparative experiments: one in which c-SAGAN was completely removed (i.e., no augmentation), and another where it was replaced by a classic conditional Wasserstein GAN (c-WGAN). This setup allows us to verify the necessity of self-attention-based augmentation and compare different generation strategies. As shown in Table 3, removing c-SAGAN leads to a significant drop in recognition accuracy on all tasks. For instance, the DEAP-Valence accuracy dropped from 70.82% (with c-SAGAN) to 65.11% without augmentation. Moreover, c-WGAN performed worse than both c-SAGAN and the non-augmentation baseline, highlighting the superiority of c-SAGAN in modeling complex multimodal patterns.

Next, we examined the effectiveness of adversarial learning in LCAN. As shown in Table 4, removing the adversarial training process resulted in noticeable performance degradation across all tasks. For example, on the DEAP-Arousal task, accuracy dropped from 71.34% to 62.09%. Similarly, performance decreased by over 8% on the WESAD dataset. These results demonstrate the crucial role of LCAN in promoting cross-modal and cross-subject feature alignment and enhancing generalization.

Finally, we investigated the structural design of the CT module by varying the number of Transformer layers from 1 to 5 and introducing a baseline (CT-0) in which the Transformer was completely removed. As illustrated in Table 5, model performance improves as the number of layers increases from 1 to 4, peaking at 70.82% (Valence) and 71.34% (Arousal). Adding a fifth layer slightly reduces performance, indicating potential overfitting. The CT-0 variant shows a drastic performance drop to 52.45% (Valence) and 50.23% (Arousal), confirming that the Transformer is indispensable for effectively modeling cross-modal dependencies.

In summary, the ablation study confirms that all three modules—CT, c-SAGAN, and LCAN—play essential roles in the CT-ELCAN framework. Each component contributes to performance gains in different but complementary ways. The Transformer captures complex inter-modal relationships, c-SAGAN enriches the training distribution with label-aware samples, and LCAN enforces feature invariance through adversarial training. Together, these modules collectively elevate the model’s accuracy, robustness, and generalizability in multimodal emotion recognition tasks.

### 5.4. Other Impact Factors

In the preceding experiments, we focused on multimodal emotion recognition tasks using the DEAP and WESAD datasets under ideal conditions with all four physiological modalities available. However, in real-world applications, collecting all modalities simultaneously is often impractical due to equipment limitations and high acquisition costs. Moreover, environmental factors during signal collection—such as noise interference or electrode displacement—may severely affect signal quality. To evaluate the practical robustness of the proposed CT-ELCAN model, we conducted additional experiments simulating two common scenarios: modality loss and signal degradation due to noise.

Using the DEAP dataset as an example, two evaluations were conducted: (1) *Modality Loss Experiments*, where each modality (EEG, EMG, EOG, and GSR) was individually removed to observe the resulting impact on recognition performance; and (2) *Noise Interference Experiments*, where Gaussian noise was artificially added to the full multimodal input to test the model’s tolerance to signal corruption.

As shown in Figure 6 and Figure 7, removing modalities resulted in performance degradation of varying degrees for both datasets. On DEAP, removing EEG caused the largest accuracy drop—valence classification decreased to 51.18% (from 70.82%), and arousal dropped to 52.69% (from 71.34%). This confirms EEG’s primary contribution, as it encodes the most emotion-relevant information among the modalities. In contrast, the removal of EMG, EOG, or GSR led to moderate accuracy reductions: the four-category task dropped to 58.36%, 58.92%, and 55.89%, respectively, compared to 63.07% with all modalities.

Noise robustness was also verified, as shown in Figure 8. Even after introducing Gaussian noise, the model retained relatively high accuracy on DEAP, with valence and arousal scores dropping only slightly to 67.08% and 66.88%, respectively—indicating just a 3–4% decline. Likewise, the quaternary classification task on DEAP and the three-class task on WESAD saw accuracy reductions of approximately 3% and 2%, respectively. These minor performance losses highlight CT-ELCAN’s strong resilience, which can be attributed to the internal data augmentation provided by c-SAGAN.

In summary, while the CT-ELCAN model exhibits sensitivity to modality absence—particularly EEG—it remains stable and effective under partial signal loss and noise interference. These findings confirm the model’s robustness and practical value, and they provide useful guidance for deployment in real-world environments where data incompleteness and signal artifacts are common.

## 6. Discussion

Although CT-ELCAN achieves strong performance in cross-subject emotion recognition, its applicability remains subject to several objective constraints.

First, the model relies on uniformly labeled data for supervised training, whereas emotional responses are inherently subjective. Different individuals may react differently to the same stimulus, and a single label may not accurately capture the true emotional state of each subject. This label uncertainty can limit the model’s ability to generalize across individuals. In our future work, we plan to incorporate label confidence modeling, soft supervision, or multi-annotator fusion strategies to improve robustness to subjective variability.

Second, the current model assumes that multimodal physiological signals are complete and temporally aligned. In real-world applications, however, signal dropout, modality loss, or desynchronization are common. The model’s robustness under such conditions remains limited. To address this, we intend to explore modality completion networks, uncertainty-aware fusion, and adaptive inference strategies based on available modalities.

Finally, while CT-ELCAN currently focuses on physiological modalities, we aim to extend our framework by integrating non-physiological signals such as speech, facial expressions, and text. These heterogeneous sources may enhance affective perception, but also introduce challenges such as modality alignment, temporal synchronization, and semantic fusion. Addressing these issues will be an important direction in our future research.

## 7. Conclusions

In this study, we proposed a novel framework for multimodal emotion recognition, termed **CT-ELCAN**, which integrates a Cross-modal Transformer (CT) with an Enhanced Learning-Classifying Adversarial Network (ELCAN). This design effectively addresses key challenges in multimodal physiological signal analysis, including inter-modality heterogeneity, subject variability, and training instability. Extensive experiments on two benchmark datasets, DEAP and WESAD, demonstrate that CT-ELCAN achieves substantial improvements in cross-subject recognition accuracy—surpassing existing state-of-the-art methods by approximately 7% and 5%, respectively. In addition, the model exhibits strong robustness and stability under challenging conditions such as modality loss and signal degradation, validating its practical applicability. Overall, the experimental results confirm the effectiveness of CT-ELCAN in extracting modality-invariant and subject-invariant representations, offering a promising solution for robust and generalizable multimodal emotion recognition.

## Figures and Tables

**Figure 1 bioengineering-12-00528-f001:**
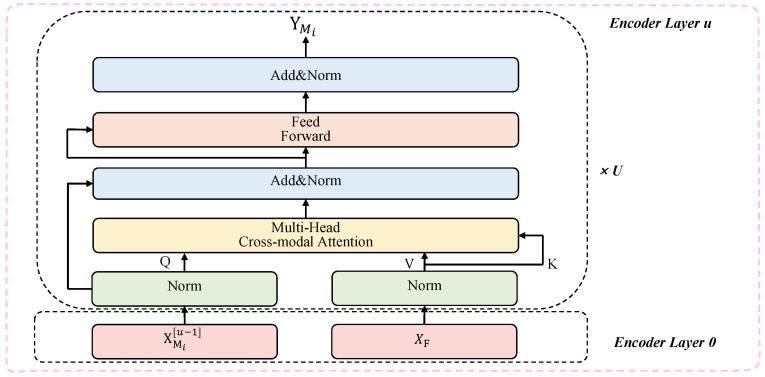
The structure of Cross-modal Transformer.

**Figure 2 bioengineering-12-00528-f002:**
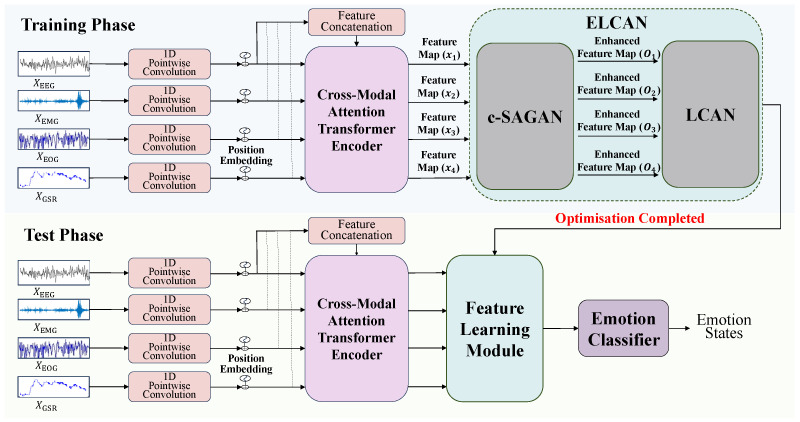
Overall architecture of the proposed CT-ELCAN framework. The system is composed of a cross-modal Transformer for feature extraction and an adversarial learning module (ELCAN), which includes LCAN and c-SAGAN submodules.

**Figure 3 bioengineering-12-00528-f003:**
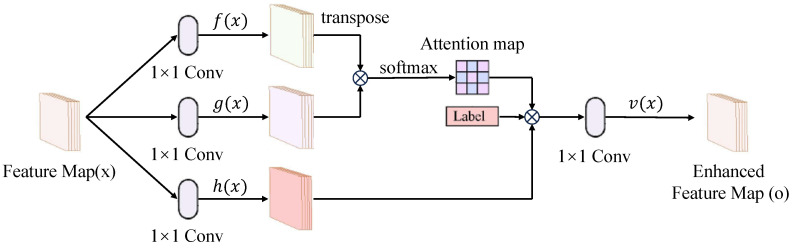
The structure of c-SAGAN submodule.

**Figure 4 bioengineering-12-00528-f004:**
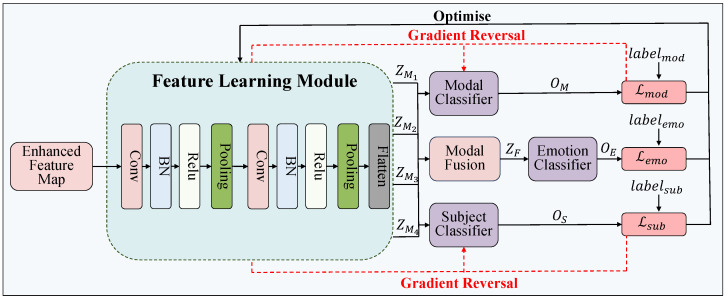
The structure of LCAN.

**Figure 5 bioengineering-12-00528-f005:**
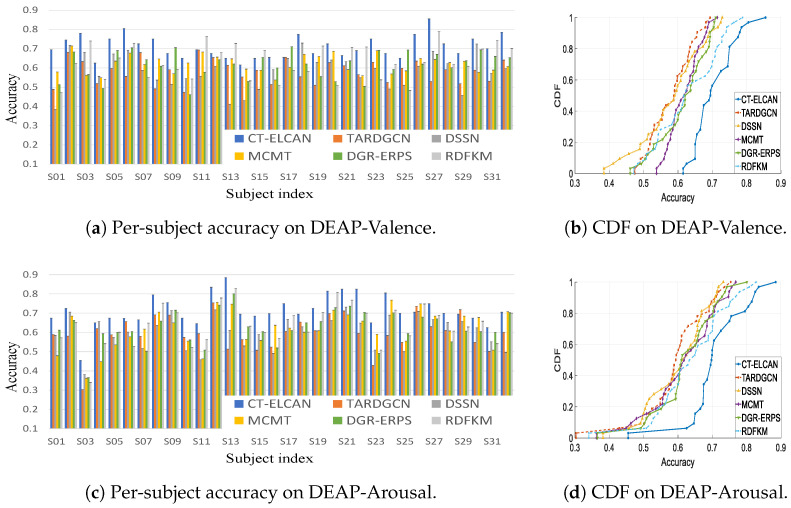
Subject-wise accuracy and CDF results of LOSO cross-validation on DEAP and WESAD datasets.

**Figure 6 bioengineering-12-00528-f006:**
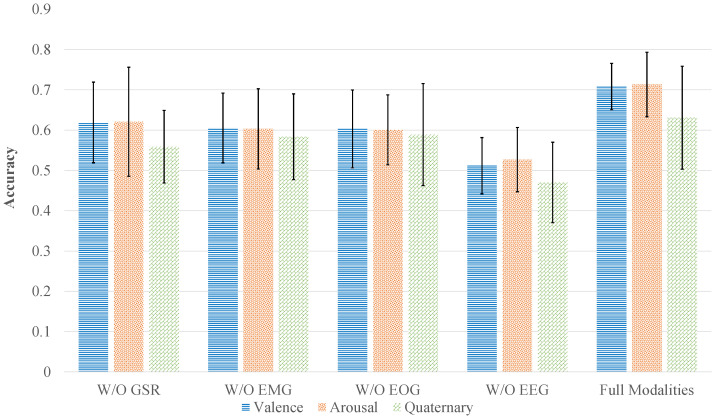
Impact of modality removal on CT-ELCAN’s performance using the DEAP dataset.

**Figure 7 bioengineering-12-00528-f007:**
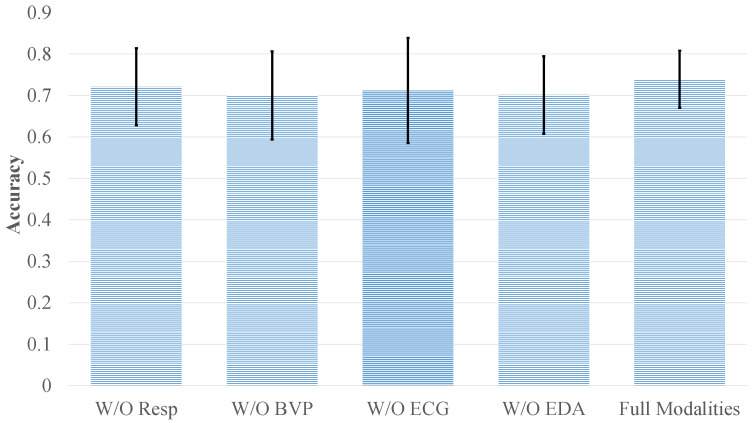
Impact of modality removal on CT-ELCAN’s performance using the WESAD dataset.

**Figure 8 bioengineering-12-00528-f008:**
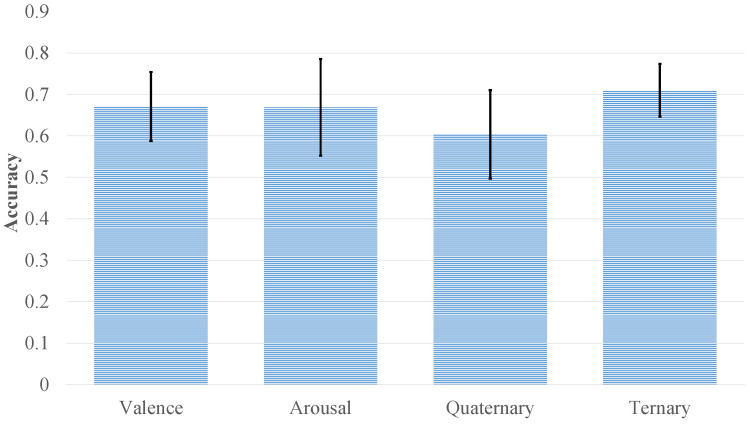
Impact of Gaussian noise on model performance across multiple classification tasks.

**Table 1 bioengineering-12-00528-t001:** Model parameter settings.

Parameter	Value
Optimizer	Adam
Learning rate	$10^{-4}$
Batch size	128
Epochs	50
Dropout ratio	0.1
c-SAGAN learning rate	$10^{-4}$
Transformer layers	4
Transformer heads	4

**Table 2 bioengineering-12-00528-t002:** LOSO accuracy results for DEAP and WESAD using different methods.

Method	DEAP-Valence	DEAP-Arousal	DEAP-Quaternary	WESAD
CNN	50.37/4.63	51.02/2.84	39.85/3.15	53.71/5.92
LSTM	49.82/5.11	50.67/7.33	43.26/3.07	52.94/6.45
GRU	51.15/2.89	51.33/6.95	44.03/2.82	54.12/5.03
Transformer	56.29/8.45	57.73/9.88	50.12/5.33	59.64/7.69
TARDGCN	58.24/5.87	59.71/9.14	55.46/8.41	-
DSSN	58.32/8.16	60.22/8.66	57.77/9.45	-
MCMT	61.96/4.28	61.80/8.45	59.28/9.64	65.78/8.10
DGR-ERPS	61.74/6.37	62.40/7.82	60.44/9.53	66.34/7.14
RDFKM	63.06/8.59	64.00/8.88	61.69/10.03	68.73/6.70
CT-ELCAN (ours)	**70.82/5.77**	**71.34/8.01**	**63.07/12.78**	**73.93/6.89**

**Table 3 bioengineering-12-00528-t003:** Ablation results: impact of different data augmentation strategies.

Method	DEAP-Valence	DEAP-Arousal	DEAP-Quaternary	WESAD
None	65.11/8.53	66.53/12.03	58.83/11.92	69.93/9.11
c-WGAN	62.95/9.34	63.10/5.96	60.29/10.20	65.17/10.36
c-SAGAN	70.82/5.77	71.34/8.01	63.07/12.78	73.93/6.89

**Table 4 bioengineering-12-00528-t004:** Ablation results: effect of adversarial learning in LCAN.

Method	DEAP-Valence	DEAP-Arousal	DEAP-Quaternary	WESAD
Without	60.26/7.02	62.09/11.93	55.02/9.23	64.90/6.19
With	70.82/5.77	71.34/8.01	63.07/12.78	73.93/6.89

**Table 5 bioengineering-12-00528-t005:** Ablation results: performance with different Transformer depths.

Method	DEAP-Valence	DEAP-Arousal	DEAP-Quaternary	WESAD
CT-0	52.45/6.17	50.23/8.14	42.37/10.92	61.32/7.08
CT-1	69.12/6.08	69.87/8.10	60.98/11.85	71.80/6.05
CT-2	69.56/6.03	70.12/8.07	61.12/12.02	72.11/7.22
CT-3	70.01/5.97	70.45/8.03	61.58/12.28	72.25/6.95
CT-4	70.82/5.77	71.34/8.01	63.07/12.78	73.93/6.89
CT-5	70.39/5.90	71.01/8.00	62.47/9.80	73.55/5.90

## Data Availability

The datasets used in this study are publicly available. The DEAP dataset can be accessed at http://www.eecs.qmul.ac.uk/mmv/datasets/deap/ (accessed on 15 April 2025), and the WESAD dataset is available at https://archive.ics.uci.edu/ml/datasets/WESAD, (accessed on 15 April 2025). No new data were generated in this study.

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
