# Peer review of "Cross-Subject Emotion Recognition with CT-ELCAN: Leveraging Cross-Modal Transformer and Enhanced Learning-Classify Adversarial Networkâ€"

_bioengineering, 2025, doi:10.3390/bioengineering12050528_

Round 1
Reviewer 1 Report
Comments and Suggestions for Authors
This paper proposes CT-ELCAN, a novel framework combining a cross-modal Transformer with an Enhanced Learning-Classifying Adversarial Network to address modality heterogeneity and inter-subject variability in multimodal physiological emotion recognition. By focusing on aligning modality- and subject-invariant emotional representations rather than direct signal fusion, CT-ELCAN significantly improves generalization and robustness. Experiments on DEAP and WESAD datasets show accuracy gains of approximately 7% and 5% over state-of-the-art methods. This paper presents an interesting approach; however, several critical issues remain unresolved:
- The Contributions section should highlight the algorithmic model’s theoretical or application-driven innovations, rather than summarizing conclusions. For example, the third listed contribution should be deleted.
- The literature review in Chapter 2 is not comprehensive enough. It is necessary to include more references, particularly from relevant studies published in this journal within the past three years.
- The criteria for setting the model parameters are not explained. A justification for the parameter values provided in Table 1 should be clearly presented.
- Although the proportion of newly proposed models in the comparative study is appropriate, it is also important to compare the proposed method against classic models such as CNN, GRU, and Transformer to better contextualize its performance.
- The conclusion section should include a discussion of the limitations of the proposed model to provide a balanced evaluation of the work.
Author Response
Comments 1: The Contributions section should highlight the algorithmic model’s theoretical or application-driven innovations, rather than summarizing conclusions. For example, the third listed contribution should be deleted.
Response 1: Thank you for the helpful suggestion. We have revised the contributions to better emphasize the technical innovations of our work. Specifically, the third listed contribution has been removed as recommended. The change can be found in page 3, line 98.
Comments 2: The literature review in Chapter 2 is not comprehensive enough. It is necessary to include more references, particularly from relevant studies published in this journal within the past three years.
Response 2: We agree with the reviewer’s comment. In Section 2, we have added three recent and representative references published in this journal to enhance the breadth and relevance of the literature review. These have been inserted as references [21], [22], and [23]. The change can be found in page 3, line 126 and page4, line 130.
Comments 3: The criteria for setting the model parameters are not explained. A justification for the parameter values provided in Table 1 should be clearly presented.
Response 3: Thank you for pointing this out. In Section 5.1, we now provide an explanation of the model parameter selection process. The parameter values in Table 1 were determined through comprehensive grid search experiments to ensure optimal performance.
The change can be found in page 11, line 359 to 366.
The corresponding revision is as follows:
“For model training, we adopted the Adam optimizer [33] with a fixed learning rate of , and applied gradient clipping when the gradient norm exceeded 10 to ensure training stability. All models were trained for 50 epochs with a batch size of 128 using the PyTorch framework [34], executed on a GeForce GTX 1080Ti GPU. Task-level parallelism was optimized to accommodate hardware constraints. To further improve performance across training rounds, we employed a grid search strategy to comprehensively explore the hyperparameter space and identify the optimal configuration that maximizes model performance. The final parameter settings are summarized in Table 1.”
Comments 4: Although the proportion of newly proposed models in the comparative study is appropriate, it is also important to compare the proposed method against classic models such as CNN, GRU, and Transformer to better contextualize its performance.
Response 4: We appreciate this suggestion. We have expanded the experimental comparison in Section 5.2 to include baseline results from classical models including CNN, LTSM, GRU, and Transformer. These models were trained and evaluated under the same conditions as CT-ELCAN for a fair comparison.
The change can be found in page 11, line 370 to page 12, line381.
The corresponding revision is as follows:
“
To evaluate the effectiveness and generalization ability of the proposed CT-ELCAN framework, we conducted comprehensive cross-subject comparative experiments under the Leave-One-Subject-Out (LOSO) cross-validation setting.
We first compared CT-ELCAN with several widely adopted deep learning models, including CNN, LSTM, GRU, and Transformer. As shown in the upper part of Table 2, these classical architectures exhibit limited ability to handle modality heterogeneity and inter-subject variability. Among them, the Transformer model achieves the highest accuracy, benefiting from its attention mechanism; however, it still falls short in terms of generalization.
To further assess performance, we compared CT-ELCAN against recent state-of-the-art methods, including single-modal EEG-based models such as TARDGCN [35] and DSSN [36], as well as multimodal physiological signal-based models such as RDFKM [37], MCMT [38], and DGR-ERPS [39]. All models were trained and evaluated under identical experimental conditions to ensure fairness and comparability. The corresponding results are summarized in the lower part of Table 2.
As shown in Table 2, CT-ELCAN consistently outperforms all competing methods across four classification tasks. On the DEAP dataset, it achieves average accuracies of 70.82% and 71.34% for valence and arousal classification, respectively, surpassing the best-performing baseline RDFKM by over 7%. In the more challenging four-class classification task on DEAP, it also achieves the highest accuracy of 63.07%. On the WESAD dataset, CT-ELCAN reaches an accuracy of 73.93%, outperforming RDFKM by over 5%.
”
Comments 5: The conclusion section should include a discussion of the limitations of the proposed model to provide a balanced evaluation of the work.
Response 5: As suggested, we have added a new Section 6 entitled “Discussion,” which explicitly discusses the main limitations of CT-ELCAN. This includes the subjectivity of emotion labels, dependence on modality completeness, and the lack of non-physiological signal integration. We also briefly outline potential solutions and future directions.
The change can be found in page 16, line 473 to page 17, line493.
The corresponding revision is as follows:
“
- Discussion
Although CT-ELCAN achieves strong performance in cross-subject emotion recognition, its applicability remains subject to several objective constraints.
First, the model relies on uniformly labeled data for supervised training, whereas emotional responses are inherently subjective. Different individuals may react differently to the same stimulus, and a single label may not accurately capture the true emotional state of each subject. This label uncertainty can limit the model’s ability to generalize across individuals. In our future work, we plan to incorporate label confidence modeling, soft supervision, or multi-annotator fusion strategies to improve robustness to subjective variability.
Second, the current model assumes that multimodal physiological signals are complete and temporally aligned. In real-world applications, however, signal dropout, modality loss, or desynchronization are common. The model’s robustness under such conditions remains limited. To address this, we intend to explore modality completion networks, uncertainty-aware fusion, and adaptive inference strategies based on available modalities.
Finally, while CT-ELCAN currently focuses on physiological modalities, we aim to extend our framework by integrating non-physiological signals such as speech, facial expressions, and text. These heterogeneous sources may enhance affective perception, but also introduce challenges such as modality alignment, temporal synchronization, and semantic fusion. Addressing these issues will be an important direction in our future research.
”

Reviewer 2 Report
Comments and Suggestions for Authors
i am grateful for the opportunity to review your manuscript. the subject that your work addresses is needful and timely, and you have done a robust job of setting up and justifying the context, gap, and need for the specific line of inquiry you have embarked on, namely an accurate reading of human emotion in real-world settings. i feel confident that you recognize the limitations in existing attempts at interpreting and deducing complex and multi-dimensional information such as inter-subject and culturally-variant emotion recognition.
the study itself is sufficiently distinct from the work you shared at last year's BICS conference, and your description of and justification for the design of the overall algorithmic architecture is clear. i also appreciate your use of CT-ELCAN on both the DEAP and WESAD datasets as this facilitates transparency in methodology and interpretation.
i am happy to endorse your manuscript for subsequent stages in the publication process. as a minor point, the positioning of Figure 2 (especially) within this iteration of the body text is 'off', as it is positioned within section 3, but (rightly) referred to in section 4 when you introduce the methodology. similar instances happen with Figures 3 and 4, but this is less problematic because the mis-placements still occur within the same section of the overall structure of the text. do consult the editors if the present placement of Figure 2 in section 3 (rather than in section 4, when it is actually referred to) is acceptable.
Author Response
Comments 1: i am grateful for the opportunity to review your manuscript. the subject that your work addresses is needful and timely, and you have done a robust job of setting up and justifying the context, gap, and need for the specific line of inquiry you have embarked on, namely an accurate reading of human emotion in real-world settings. i feel confident that you recognize the limitations in existing attempts at interpreting and deducing complex and multi-dimensional information such as inter-subject and culturally-variant emotion recognition.
the study itself is sufficiently distinct from the work you shared at last year's BICS conference, and your description of and justification for the design of the overall algorithmic architecture is clear. i also appreciate your use of CT-ELCAN on both the DEAP and WESAD datasets as this facilitates transparency in methodology and interpretation.
i am happy to endorse your manuscript for subsequent stages in the publication process. as a minor point, the positioning of Figure 2 (especially) within this iteration of the body text is 'off', as it is positioned within section 3, but (rightly) referred to in section 4 when you introduce the methodology. similar instances happen with Figures 3 and 4, but this is less problematic because the mis-placements still occur within the same section of the overall structure of the text. do consult the editors if the present placement of Figure 2 in section 3 (rather than in section 4, when it is actually referred to) is acceptable.
Response 1: We sincerely thank the reviewer for the supportive and encouraging comments. We greatly appreciate your recognition of the importance and originality of our work, as well as the clarity of our algorithmic design and the application across multiple datasets.
Regarding the comment about the positioning of Figure 2: we acknowledge that Figure 2 is referenced in Section 4 while currently positioned at the end of Section 3. However, we have chosen to keep this placement for the sake of layout consistency and visual clarity, as all figures in the manuscript are top-aligned and designed to appear on the same page as their first mention. Since Figure 2 and Section 4 appear on the same page, we believe this placement still facilitates reader comprehension without disrupting the logical flow.
Therefore, we respectfully decided not to change the figure placement. Should the editors deem it necessary to adjust, we will be happy to comply with the journal’s formatting guidelines.

Round 2
Reviewer 1 Report
Comments and Suggestions for Authors
The revised manuscript has addressed key concerns with substantial improvements. I recommend acceptance in its current form.